

# Replication study: androgen receptor splice variants determine taxane sensitivity in prostate cancer

Xiaochuan Shan[1], Gwenn Danet-Desnoyers[1], Fraser Aird[2], Irawati Kandela[2], Rachel Tsui[3], Nicole Perfito[3] and Elizabeth Iorns[3]

[1] Stem Cell and Xenograft Core, Perelman School of Medicine, Philadelphia, PA, USA
[2] Developmental Therapeutics Core, Northwestern University, Evanston, IL, USA
[3] Science Exchange and The Prostate Cancer Foundation–Movember Foundation Reproducibility Initiative, Palo Alto, CA, USA

## ABSTRACT

In 2015, as part of the Prostate Cancer Foundation–Movember Foundation Reproducibility Initiative, we published a Registered Report (*Shan et al., 2015*) that described how we intended to replicate selected experiments from the paper "Androgen Receptor Splice Variants Determine Taxane Sensitivity in Prostate Cancer" (*Thadani-Mulero et al., 2014*). Here we report the results of those experiments. Growth of tumor xenografts from two prostate cancer xenograft lines, LuCaP 86.2, which expresses wild-type androgen receptor (AR) and AR variant 567, and LuCaP 23.1, which expresses wild-type AR and AR variant 7, were not affected by docetaxel treatment. The LuCaP 23.1 tumor xenografts grew slower than in the original study. This result is different from the original study, which reported significant reduction of tumor growth in the LuCaP 86.2. Furthermore, we were unable to detect ARv7 in the LuCaP 23.1, although we used the antibody as stated in the original study and ensured that it was detecting ARv7 via a known positive control (22rv1, *Hörnberg et al., 2011*). Finally, we report a meta-analysis of the result.

## INTRODUCTION

The Prostate Cancer Foundation–Movember Foundation Reproducibility Initiative is a collaboration between Science Exchange and the Prostate Cancer Foundation aiming to replicate key experimental results from high impact papers published in the prostate cancer field. For each of these papers, a Registered Report detailing the proposed experimental designs and protocols for the replications was peer reviewed and published prior to data collection. The present paper is a Replication Study that reports the results of the replication experiments described in the Registered Report (*Shan et al., 2015*) for a paper by *Thadani-Mulero et al. (2014)* and uses a number of approaches to compare the outcomes of the original experiments and the replications.

Thadani-Mulero and colleagues reported that ARv567 and ARv7 responded differently to docetaxel in their 2014 Cancer Research paper. They found that the ARv567 variant was

Corresponding authors
Rachel Tsui,
rachel.tsui@scienceexchange.com
Elizabeth Iorns,
elizabeth@scienceexchange.com

heavily associated with microtubules, while the ARv7 variant was not, which resulted in a significant decrease of nuclear accumulation of ARv567 but not ARv7 with docetaxel treatment. They performed a xenograft tumor growth assay on two different prostate cancer cell lines: LuCaP 86.2, which expresses wild-type androgen receptor (AR) and ARv567, and LuCaP 23.1, which expresses wild-type AR and ARv7. They reported that docetaxel treatment reduced tumor growth significantly in tumors derived from LuCaP 86.2 cells, while tumors derived from LuCaP 23.1 did not show tumor growth reduction, suggesting that docetaxel affects the AR splice variants differently.

In a Registered Report, Shan and colleagues described the experiments to be replicated for key results of the Thadani-Mulero paper (Figs. 6A and 6B) and summarized the current evidence for these findings (*Shan et al., 2015*). Since the publication of the Registered Report, there have been no additional studies examining the AR signaling profile of LuCaP 86.2 and 23.1 xenograft tumors. However, a study by *Mang et al. (2015)* suggests that high CEP57 expression in prostate cancer tumors is associated with mitotic impairment and, thereby, less aggressive tumor behavior and that the CEP57-induced microtubule stabilization does not affect AR nuclear translocation. This suggests that docetaxel may have other modes of action independent of AR transport inhibition.

## MATERIALS AND METHODS

As described in the Registered Report (*Shan et al., 2015*), we attempted a replication of the experiments reported in Fig. 6A of *Thadani-Mulero et al. (2014)*. A detailed description of all protocols can be found in the Registered Report (*Shan et al., 2015*). Additional detailed experimental notes, data, and analysis are available on https://osf.io/gkd2u/.

*Protocol 1: response of xenograft tumors derived from LuCaP 86.2 and LuCaP 23.1 prostate cancer cells to treatment with docetaxel*

LuCaP 86.2 and 23.1 tumor tissues were shipped fresh and kindly shared by the original authors and subcutaneously implanted into six-to-eight week old mice homozygous for the severe combined immune deficiency (SCID) spontaneous mutation on the right flank-shoulder area. Tissue was tested for pathogens prior to implantation (https://osf.io/t9cm4/). Male CB17 SCID mice were obtained from Charles River (Strain Code 236). This protocol was approved by the Institutional Animal Care and Use Committee (Animal Welfare Assurance #A3079-01 for Protocol #803506).

Tumors were then grown to 100 mm$^3$ prior to start of docetaxel treatment. While the original publication state that treatment began at 200 mm$^3$ in the "Materials and Methods," the original data in Fig. 6A and Fig. S4 of Thadani-Mulero et al. indicate that treatment began at 100 mm$^3$, which is the protocol we followed as per the Registered Report. Once tumors reached 100 mm$^3$, mice were randomly assigned to treatment group or control group. A total of 19 of the LuCaP 23.1 xenograft mice developed tumors and were enrolled in treatment or control experiments (10 for docetaxel treatment and nine for control). A total of 26 of the LuCaP 86.2 xenograft mice developed tumors and were enrolled in treatment or control experiments (13 for docetaxel and 13 for

control). Docetaxel was obtained from LC Laboratories (D-1000) and treated mice received 10 mg/kg docetaxel in 400 μL 13% ethanol/0.9% NaCl per injection.

Tumor growth was measured twice weekly and treatment continued for eight weeks with docetaxel injections at weeks 1, 3, 5, 7. Mice would be euthanized when they displayed one or more of the following conditions: (1) tumor volume exceeding 1,000 mm³, (2) >20% body weight loss, or (3) animals becoming compromised. At week 8, mice were sacrificed and tumor tissues were harvested for analysis.

*Protocol 2: Western blot analysis confirming expression of AR truncation mutants in xenograft tumor tissue*

Three uninjected LuCaP 86.2 and three uninjected LuCaP 23.1 tumor tissues were harvested from the untreated groups and snap-frozen. The samples were then lysed in TNES buffer, homogenized, and centrifuged at 13,000 rpm. The supernatant was then diluted 1:1 and total protein concentration measured using the Bradford Protein Assay.

Negative Control: The PC-3 cell line is one of the classically used prostate cancer cell lines that is characterized as not containing detectable AR levels (*Mitchell et al., 2000*).

Positive Control: Cell Signaling Technologies provided the LNCaP cell line as a positive control for AR detection, particularly wild-type AR. This was also seen in a study conducted by *Cronauer et al. (2004)*.

Positive Control: 22rv1 is another positive control cell line for AR detection, and detects wild-type AR and variants of AR (*Hörnberg et al., 2011*).

A total of 30 μg protein per well was electrophoresed through a 10% SDS-PAGE gel (Mini-PROTEAN® TGX Precast Gel, 456-1033; Bio-Rad, Hercules, CA, USA) at 130 V/70 min in running buffer (Tris-glycine-SDS, 161-0732; Bio-Rad, Hercules, CA, USA). For the transfer, a PVDF membrane (Immuno-Blot 0.2 μm, 162-0174; Bio-Rad, Hercules, CA, USA) was first wet in methanol (34860; Sigma-Aldrich, St. Louis, MO, USA), then in transfer buffer (Tris-glycine, 161-0734 + 20% methanol; Bio-Rad, Hercules, CA, USA). Proteins were blotted onto the PVDF membrane by electrophoretic transfer at 250 mA for 1 h.

The membrane was washed in Tris-buffered saline (170-6435; Bio-Rad, Hercules, CA, USA) with 0.05% Tween-20 (170-5017; Bio-Rad, Hercules, CA, USA; TBS-T) for 10 min at room temperature, then blocked in 5% non-fat dried milk (NFDM, M7409; Sigma-Aldrich, St. Louis, MO, USA)/TBS-T for 1 h at room temperature. The membrane was washed three times for 5 min each at room temperature in TBS-T and cut into an upper section and a lower section at around 60–65 kDa.

The upper section was incubated with primary antibody diluted in 5% NFDM/TBS-T overnight at 4 °C. Each sample was probed with the primary antibodies for the following targets at the following dilutions:

- AR(N-20) (sc-816; Santa Cruz Biotechnology, Santa Cruz, CA, USA) 1:200
- AR-V7 (AG10008; Precision Antibody, Columbia, MD, USA) 1:1,000
- AR-V5,6,7es (ab200827; Abcam, Cambridge, UK) 1:1,000
- AR(D6F11) (5153; Cell Signaling Technology, Danvers, MA, USA) 1:2,000

The membrane was washed three times for 5 min each at room temperature in TBS-T, and incubated with HRP-conjugated secondary antibody diluted in 5% NFDM/TBS-T for 1 h at room temperature. Antibody dilutions were:

- Anti-rabbit IgG (sc-2030; Santa Cruz Biotechnology, Santa Cruz, CA, USA) 1:2,000 for AR(D6F11), AR-V5,6,7es, and AR(N-20)
- Anti-mouse IgG (172-1011; Bio-Rad, Hercules, CA, USA) 1:10,000 for AR-V7

The lower section was incubated with horseradish peroxidase (HRP) substrate-conjugated mouse β-actin antibody (A3854; Sigma-Aldrich, St. Louis MO, USA) diluted at 1:25,000 in 5% NFDM/TBS-T for 1 h at room temperature.

After primary and secondary antibody probing, membranes were washed three times for 5 min each at room temperature in TBS-T, and once for 5 min at room temperature in phosphate-buffered saline. HRP signal was detected by enhanced chemiluminescence using the Supersignal West Pico PLUS kit (34577; ThermoFisher Scientific, Waltham, MA, USA), followed by exposure to autoradiographic film.

## Statistical analysis

Statistical analysis was performed with R software (*R Core Team, 2014*), version 3.2.3 and Prism 7.0d. All data csv files and analysis scripts are available at https://osf.io/kcyhs/. Confirmatory statistical analysis was pre-registered before the experimental work began as outlined in the Registered Report (*Shan et al., 2015*). Data were checked to ensure assumptions of statistical tests were met. A meta-analysis of a common original and replication effect size was performed using a random effects model and the metafor R package (available at https://osf.io/xcaj2/) (*Viechtbauer, 2010*). The original study data were extracted a priori from the published figures by determining the mean and upper/lower error values for each data point. The extracted data were published in the Registered Report (*Shan et al., 2015*) and were used in the power calculations to determine the sample sizes for this study.

## Deviations from the Registered Report

*Protocol 1: response of xenograft tumors derived from LuCaP 86.2 and LuCaP 23.1 prostate cancer cells to treatment with docetaxel*

Based on the Registered Report, there should have been a total of 24 mice injected with LuCaP 23.1 and 24 mice with LuCaP 86.2 aiming for 11 mice per treatment group. In this replication study, we only injected 22 mice with LuCaP 23.1 due to the limited amount of LuCaP 23.1 tumor material we received from the original authors. For LuCaP 86.2, we received the tumor material that was more than enough for 24 mice, so we injected a total of 28 mice. Nineteen of the LuCaP 23.1 mice developed tumors and were enrolled in the treatments (nine for control and 10 for docetaxel), while 26 of the LuCaP 86.2 mice developed tumors and were enrolled in the treatments (13 for control and 13 for docetaxel).

*Protocol 2: Western blot analysis confirming expression of AR truncation mutants in xenograft tumor tissue*

An in-house Western blot protocol was used, replacing the Li-COR Odyssey detection used by the original lab. An additional primary antibody for the androgen receptor, AR (D6F11) (5153; Cell Signaling Technology, Danvers, MA, USA), was used to confirm androgen receptor detection.

## RESULTS AND DISCUSSION

### AR variant expression profile of LuCaP 86.2 and LuCaP 23.1 prostate cancer cells

In order to confirm the expression profile of the AR splice variants of each xenograft tissue, we performed western blot analysis of protein expression (Replication Fig. 1, https://osf.io/z3bva/, with raw data files found within the corresponding folder in https://osf.io/vdqcw/).

The original study suggested that the differing AR variant profile of the LuCaP xenograft tumors caused the differences in docetaxel sensitivity, specifically that expression of the ARv567 was responsible. The original authors found that LuCaP 23.1 expressed both wild-type AR and ARv7, while LuCap 86.2 expressed wild-type AR and ARv567 (*Thadani-Mulero et al., 2014*, Fig. 6b). We found that while both LuCaP 23.1 and 86.2 express wild-type AR, in agreement with the original study, neither xenograft tumors expressed ARv7. LuCap 86.2 did express ARv567, similar to the original study. The original study used the Santa Cruz N20 antibody to visualize wild-type AR and was raised against the N-terminus of the androgen receptor. Given this, we would expect the N20 antibody to bind all variants of AR in addition to the wild-type as the N-terminus is maintained in all AR variants (*Hörnberg et al., 2011*). As such, it is unclear why ARv7 is not present in the original Western blot of LuCaP 23.1. This replication study showed that the Precision Antibody against ARv7 did not detect this variant in either cell line, although the positive control sample showed a strong band at the expected ~75 kDa (Replication Fig. 1C). Given the size similarity between ARv567 and ARv7, as well as the use of the N20 antibody, which does not discriminate between variants of AR, one would expect multiple bands in Replication Fig. 1. However, this replication study suggests that the use of the N20 antibody failed to detect any AR expression, with bands at ~70 kDa and not the expected 110 kDa (wild-type AR). This failure in AR expression detection could be due to antibody lot/batch variability between the two studies.

An additional confirmation was conducted using the Cell Signaling Technologies antibody (D6F11), which was raised against the N-terminus of AR. This antibody suggested that wild-type AR and its variants (unspecified) are expressed in both of the LuCaP tumors. Given the lack of detection of ARv7 in the two cell lines, there is a possibility that other unspecified variants are expressed in LuCaP 23.1. We also tested an additional antibody from Abcam (ab200827) specific to AR v567, and confirmed that LuCaP 86.2 but not LuCaP 23.1 expressed this variant. This Abcam antibody was not used in the original study and is specific to ARv567.

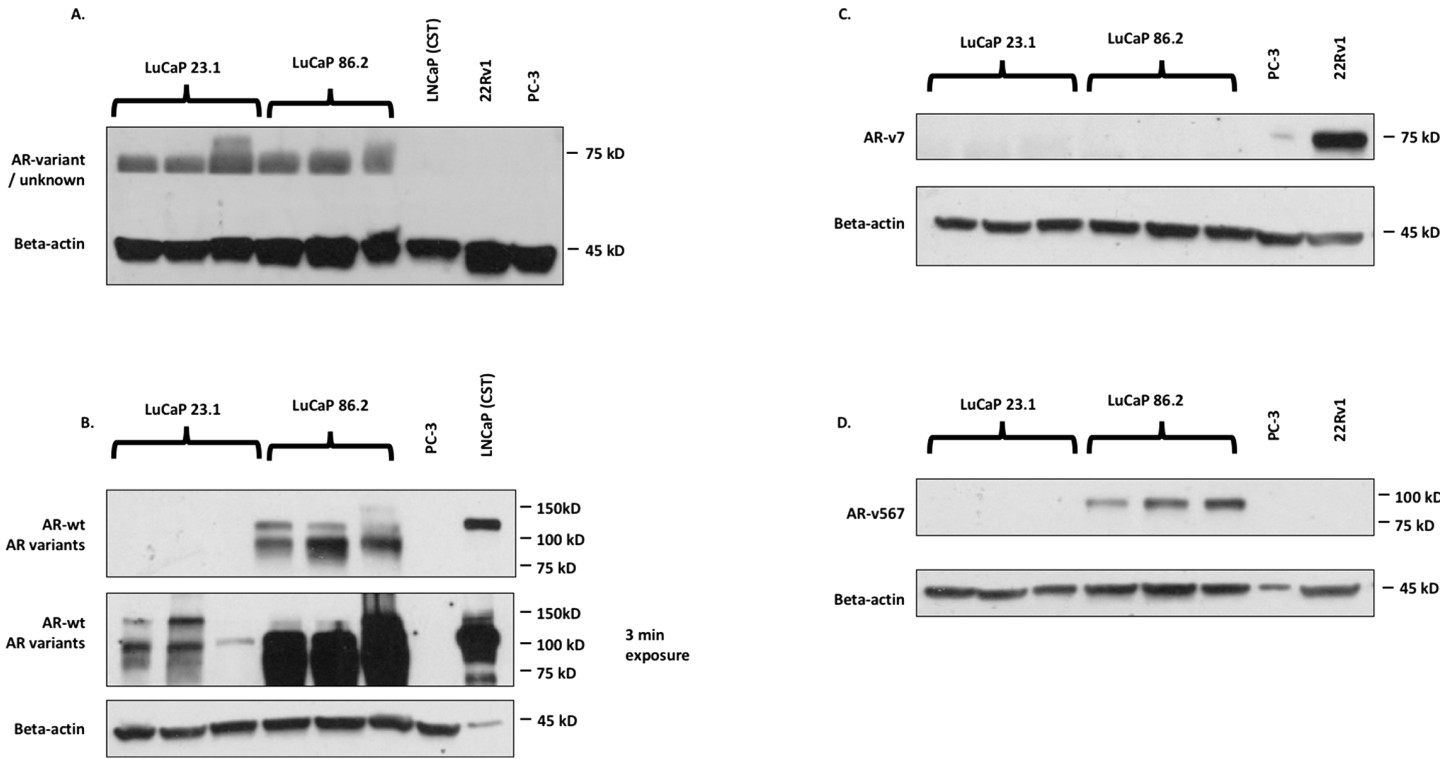

**Figure 1 Western blot images for AR detection in cell lines.** (A) N20 (SCBT), which detects AR-wt and AR variants; (B) D6F11 (Cell Signaling Technologies, Danvers, MA, USA), which detects AR-wt and AR variants; (C) AR-V7 (Precision antibody), which detects ARv7; (D) ab200827 (Abcam antibody; Abcam, Cambridge, UK), which detects ARv567.

## Response of xenograft tumors derived from LuCaP 86.2 and LuCaP 23.1 prostate cancer cells to treatment with docetaxel

This experiment is a replication of Fig. 6A from the original study (*Thadani-Mulero et al., 2014*). The LuCaP 86.2 and LuCaP 23.1 prostate cancer tissues were subcutaneously implanted into non-castrated SCID mice and tumor growth was monitored. The tumors grew to 100 mm$^3$ prior to the start of the docetaxel treatment. According to the Registered Report Analysis Plan, we compared tumor volumes at the end of the study after eight weeks of treatment, as in the original study. There was no effect of docetaxel treatment on either tumor type (effect of treatment: $F_{(1,30)} = 0.88$, $p = 0.36$; Replication Fig. 1). Mice with LuCaP 86.2 xenograft tumors had larger tumors by eight weeks of treatment than mice with LuCaP 23.1 xenograft tumors (effect of tumor type: $F_{(1,30)} = 23.24$, $p < 0.001$), and both types of xenograft tumors responded similarly with and without treatment (Interaction: $F_{(1,30)} = 0.08$, $p = 0.78$) (see Replication Fig. 2). It is unclear why the LuCaP 86.2 tumors were not sensitive to docetaxel treatment. Given the slow growth of the tumors, we were not able to directly demonstrate the negative control for LuCaP 23.1 tumors. We did not expect an effect of docetaxel in LuCaP 23.1 tumors, but tumors grew very slowly so did not show an effect of treatment.

Once potential cause of the slow growth of the LuCaP 23.1 xenograft tumors has been characterized previously in a study by *Rocchi et al. (2004)* in which they observed two

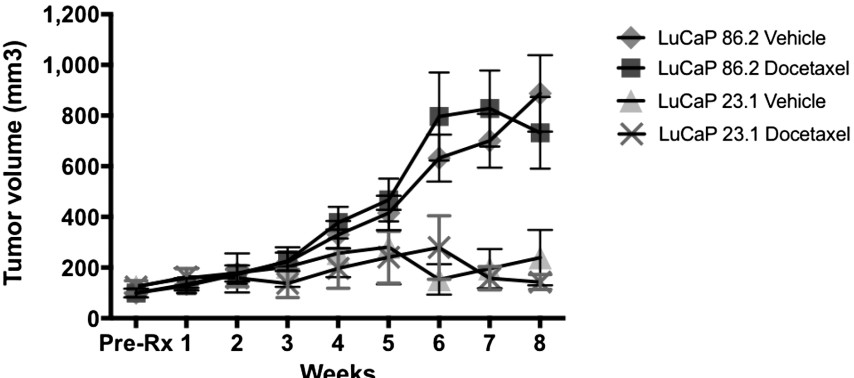

**Figure 2 Tumor growth in mice.** Effect of docetaxel on LuCaP 23.1 and 86.2 tumor growth in 45 mice. ($n = 13$ for control and docetaxel treatment in LuCaP 86.2 and $n = 9$ control, $n = 10$ docetaxel treatment in LuCaP 23.1).

distinct populations (fast-growing and slow-growing) of tumor growth in LuCaP 23.1 xenograft mice (*Rocchi et al., 2004*). According to the Registered Report Analysis Plan, the current study only measured tumor growth for eight weeks as did the original study, but does not confirm docetaxel sensitivity of LuCaP 86.2 tumor xenografts over a longer time period.

According to the Registered Report Analysis Plan, we also calculated the area under the curve to compare growth over the entire eight week period. These results were the same as the comparison at eight weeks. There was no effect of docetaxel treatment on either tumor type (effect of treatment: $F_{(1,41)} = 0.043$, $p = 0.84$; Replication Fig. 1). Mice with LuCaP 86.2 xenograft tumors had larger tumors by eight weeks of treatment than mice with LuCaP 23.1 xenograft tumors (effect of tumor type: $F_{(1,41)} = 8.75$, $p = 0.005$), and both types of xenograft tumors responded similarly (interaction: $F_{(1,41)} = 0.109$, $p = 0.74$).

In order to verify that the tumor cells used in the replication matched the original study, we analyzed the Short Tandem Repeat (STR) profile of the LuCaP prostate cancer tumor cells used by the replicating lab and found that it matched the STR profile of LuCaP prostate cancer tumor cells used by the original authors (Fig. S1, https://osf.io/w5j9e/). This suggests that differences in tumor growth cannot be explained by differences in the tumor cells used in the original study and the replication.

## Meta-analyses of original and replicated effects

We performed a meta-analysis using a random-effects model to combine each of the effects described above on week 8 tumor volume comparisons as pre-specified in the confirmatory analysis plan (*Shan et al., 2015*). To provide a standardized measure of the effect, Cohen's d was calculated for the original and replication studies. Cohen's d is the standardized difference between two means using the pooled sample standard deviation.

The comparison of LuCaP prostate cancer patient-derived tumors treated with docetaxel compared to untreated resulted in $d = 2.94$, (95% CI [1.88, 3.97]) for the data

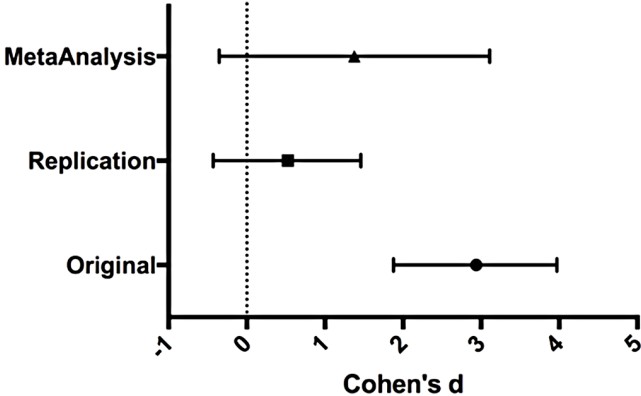

**Figure 3 Effect size (Cohen's d) and 95% CI for data.** Effect size (Cohen's d) and 95% confidence interval are presented for *Thadani-Mulero et al. (2014)*, this replication study (RP:CB), and a meta-analysis of those two effects. Sample sizes used in *Thadani-Mulero et al. (2014)* and this replication attempt are reported under the study name.     

estimated a priori from Fig. 6C of the original study (*Thadani-Mulero et al., 2014*). This compares to $d = 0.52$, (95% CI [−0.43, 1.46]) reported in this study. A meta-analysis (Replication Fig. 3) of these two effects resulted in $d = 1.38$, (95% CI [−0.36, 3.11]). The results from the replication do not show a significant effect of docetaxel on tumor growth. The point estimate of the replication effect size was not within the confidence interval of the original result and vice versa. Further, the random effects meta-analysis did not result in a statistically significant effect of docetaxel on inhibiting LuCaP derived tumors.

This attempt at assessing reproducibility is not a strong test of the original hypothesis as ARv7 was not expressed at any detectable level in the LuCaP 23.1 tumor xenografts and because these tumors grew much more slowly compared to the original study. We were also unable to see comparable data from the antibody that was used in the original study when measuring the expression of AR variants in the cell lines. Additionally, we were unable to see any effect from the docetaxel treatment from either cell line, which may have been due to this difference in growth rate between the two tumor lines, or potentially to a compromised quality of the docetaxel reagent. While a future study could include a positive quality control for the reagent, it is outside the scope of this replication study as it was not factored into the plan at the stage of the Registered Report. It was unexpected that there would be such a lack of response to the docetaxel treatment and thus a positive quality control test was not included in the Registered Report and thus not replicated here. These factors contribute to a strongly inconclusive replication attempt of the original paper as we are unable to demonstrate that the ARv7 expression resulted in the difference in docetaxel treatment.

While the difference in tumor growth could be inherent to LuCaP 23.1, it was not mentioned or seen by Thadani-Mulero in the original study. A recent paper by *Nguyen et al. (2017)* showed decreased tumor growth for both LuCaP 86.2 and 23.1 xenograft for high dose docetaxel (20 mg/kg), while only LuCaP 86.2 responded to the low dose of docetaxel (5 mg/kg). This replication study used a 10 mg/kg dose, which was

communicated by the original authors during the writing and publication of the Registered Report as the appropriate dose to use although the original study was 5 mg/kg. Nguyen and colleagues state that this low dose response is a rare exceptional responder to docetaxel, suggesting that dosage is an unlikely reason for the failure of LuCaP 86.2 to respond. Additionally, Nguyen and colleagues also found low expression levels of ARv7 mRNA relative to wild-type AR in the LuCaP 23.1, which is in line with this replication study, where we did not detect ARv7 expression in either patient derived-xenograft line. Additional in-depth studies are needed to understand whether slow vs. fast-growing tumor variant differ in AR expression and docetaxel response.

However, this replication study does provide an opportunity to understand the present evidence of these effects. Any known differences, including reagents and protocol differences, were identified prior to conducting the experimental work and described in the Registered Report (Shan et al., 2015). However, this is limited to what was obtainable from the original paper and through communication with the original authors, which means that there might be specific features of an original experimental protocol that could be crucial but unidentified at this time. Certain experimental aspects such as number of cells injected, cell line, strain and sex of mice, could be maintained, but other variables are not as easily to control for. These include the possibility that the patient derived-xenograft changes over passage, resulting in increased difficulty in obtaining consistent results. These factors can potentially influence the outcome of the study and further investigation can be facilitated by additional direct replications and transparent reporting. Some additional studies that could validate the results of the original paper but are outside the scope of this current replication study include evaluating the molecular function of the AR variants in vitro through measurement of the effects of docetaxel in AR variant overexpression systems and analysis of human data sets to investigate the association of AR variant expression and response to docetaxel (and similar microtubule-targeting drugs).

## ACKNOWLEDGEMENTS

We thank Drs. Stephen Plymate, Robert Vassella and Eva Corey for sharing protocol details before and during peer review. Dr. Vassella kindly shared tumor tissue for this study.

### Funding

The Prostate Cancer Foundation–Movember Foundation Reproducibility Initiative is funded by the Prostate Cancer Foundation and the Movember Foundation. The funders had no role in study design, data collection and analysis, decision to publish, or preparation of the manuscript.

### Grant Disclosures

The following grant information was disclosed by the authors:
Prostate Cancer Foundation.
Movember Foundation.

## Competing Interests

Elizabeth Iorns, Rachel Tsui and Nicole Perfito are employed by and hold shares in Science Exchange Inc. The experiments presented in this manuscript will be conducted by XC and GDD at the Stem Cell and Xenograft Core and by FA and IK at Developmental Therapeutics Core, which are Science Exchange labs.

## Author Contributions

- Xiaochuan Shan conceived and designed the experiments, performed the experiments, analyzed the data, contributed reagents/materials/analysis tools, prepared figures and/or tables.
- Gwenn Danet-Desnoyers conceived and designed the experiments, performed the experiments, analyzed the data, contributed reagents/materials/analysis tools.
- Fraser Aird conceived and designed the experiments, performed the experiments, analyzed the data, contributed reagents/materials/analysis tools, prepared figures and/or tables.
- Irawati Kandela conceived and designed the experiments, performed the experiments, analyzed the data, contributed reagents/materials/analysis tools.
- Rachel Tsui analyzed the data, prepared figures and/or tables, authored or reviewed drafts of the paper, approved the final draft.
- Nicole Perfito analyzed the data, prepared figures and/or tables, authored or reviewed drafts of the paper, approved the final draft.
- Elizabeth Iorns conceived and designed the experiments, analyzed the data, authored or reviewed drafts of the paper, approved the final draft.

## Animal Ethics

The following information was supplied relating to ethical approvals (i.e., approving body and any reference numbers):

This protocol was approved by the Institutional Animal Care and Use Committee (Animal Welfare Assurance #A3079-01 for Protocol #803506).

## Data Availability

Tan FE, Perfito N, Lomax J, Tsui R. 2017. "Study 3: Thadani-Mulero et al., Cancer Res 2014." Open Science Framework. https://osf.io/gkd2u.

## Supplemental Information

Supplemental information for this article can be found online at http://dx.doi.org/10.7717/peerj.4661#supplemental-information.

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
