# Peer review of "Replication study: androgen receptor splice variants determine taxane sensitivity in prostate cancer"

_PeerJ, doi:10.7717/peerj.4661_

## Round 0.1 · original submission · Major Revisions

Please address the concerns raised by the two reviewers, and provide the requested raw data/files along with your revised manuscript.

[# Staff Note: Our apologies for a delayed decision. We were waiting on a late reviewer who ultimately was unable to deliver their review #]

Reviewer 1 ·

Basic reporting

No comment.

Experimental design

No comment.

Validity of the findings

No comment.

Additional comments

1. PDX studies is notoriously hard to get consistent results, but how xenograft changes over passage is difficult to evaluate. The current report may simply reflect this reality rather than any technical issues in the original publication. There are two additional ways to validate the results of the original paper:
(1) Evaluate the molecular function of the AR variants in vitro (e.g. will overexpression of these variants change the sensitivity of the prostate cancer cells to microtubule-targeting drugs?)
(2) Analyze human data sets to check the association of AR variant expression and response to microtubule-targeting drugs.

However, these points are obviously beyond the scope of this report. If the authors don't plan to run such studies, please discuss them (including the possible reason why this xenograft change) in "Discussion".

2. Reviewer request to review the raw data of Fig. 1A (size record and growth curve of individual tumors). Please submit them.

3. Is ARv7 expressed as RNA in LuCaP23.1?

Reviewer 2 ·

Basic reporting

1. PeerJ standards: “Conclusions: Identify unresolved questions / gaps / future directions.” Please add a concluding paragraph summarizing the overall findings and differences between this study and the original study. As is the manuscript ends abruptly with no summarizing comments on the findings of the paper.
2. A PowerPoint of the N20 blot used in the manuscript can be found at https://osf.io/f39cg/. However, please provide the raw western blot for the N20 antibody used in this manuscript (the manuscript figure is not the same as the blot shown in the raw data).
3. Please note in the text of manuscript where the bands are located for the westerns. From the raw western image it is apparent that the bands with N20 are not the correct size (~70 kD when they should be ~110 kD). It is also troubling that the N20 antibody did not detect any bands for two AR positive lines (LNCaP and 22rV1).
4. Please comment on the possible discrepancy between D6F11 detecting what appears to be full-length AR and variant AR but V7 western not showing V7 (as the authors pointed out, there was also a discrepancy in original paper with this). Perhaps there are other truncation variants in the 23.1 xenograft? Also note that N20 and the Precision V7 antibodies used in the two studies are most likely different lots/batches and that could also account for differences seen (batch variability is known to occur with both of these antibodies). A recent paper by Nguyen, et al, (2017) “LuCaP Prostate Cancer Patient-Derived Xenografts Reflect the Molecular Heterogeneity of Advanced Disease and Serve as Models for Evaluating Cancer Therapeutics.” The Prostate, 7:654-671, shows decreased tumor growth for both 86.2 and 23.1 for high dose docetaxel (20 mg/kg) but only 86.2 responded to low dose docetaxel (5 mg/kg). Also this paper found low expression levels levels of AR-V7 mRNA in 23.1 (increased levels in castrate resistant 23.1).
5. Minor change: Line 178, there is no Replication Figure 1C or a graph of area under the curve shown (only 1A--tumor volume graph). If no other parts are included in Figure 1, then it should just be labeled as Figure 1.
6. The sentence on Lines 75-77 needs to be edited, “Three mice in the LuCaP 23.1 xenograft group Docetaxel was obtained from LC Laboratories (D-1000) and treated mice received 10 mg/kg docetaxel in 400 μL 13% ethanol/0.9% NaCl per injection.” The first part of the sentence should be deleted so it reads, “Docetaxel was obtained from LC Laboratories (D-1000) and treated mice received 10 mg/kg docetaxel in 400 μL 13% ethanol/0.9% NaCl per injection.”
7. Lines 74 and 75, if the control mice did not actually receive injections of the diluent alone (or vehicle) then they should be labeled simply as control mice, not vehicle mice.
8. Lines 171-173, the original study did not show docetaxel sensitivity in the 23.1 xenograft. Did the authors actually mean to say 86.2?

Experimental design

No comment

Validity of the findings

PeerJ standards: “Conclusions: Identify unresolved questions / gaps / future directions.” Please add a concluding paragraph summarizing the overall findings and differences between this study and the original study. As is the manuscript ends abruptly with no summarizing comments on the findings of the paper.

Additional comments

This is a very brief replication report of only the in vivo study performed by Thadani-Mulero, et al, 2014. In the original study the authors demonstrated a link between expression of the constitutively active androgen receptor variant ARv567es and taxane sensitivity. This replication study found the opposite, no effect of taxanes on growth of tumors with or without ARv567es. Overall the authors put together a very concise report. However, there are some places were the authors need to elaborate further or clarify points.

---

## Round 0.2 · Major Revisions

While I do agree with all comments made by Reviewer#1, I understand that most of the requested work is NOT within the scope of this Replication Study. However, I do request that the following points will be addressed in a revised manuscript.

1) In response to the comment by Reviewer#1: “Possibly because of the wide range of the data point distribution, the authors excluded the data of the fastest growing tumors from Fig. 1”, please include ALL tumor data in a revised graph and update the statistical analysis accordingly.

2) Given that none of the xenograft lines exhibited a significant response to docetaxel, the study requires a technical quality control for the batch of drug, ideally as an in vivo experiment that was conducted at the same time. If such quality control is not available, please remove the drug-treatment data, and revise the text accordingly.

3) Please revise the Discussion to state explicitly that this attempt at assessing reproducibility failed because ARv7 was not expressed at detectable level in LuPaC 23.1 tumors and because these tumors grew much more slowly compared to the original study, i.e. never reached the size reported previously. Also, discuss that (a) drug-responses could not be compared because of the difference in growth rate between the two tumor lines, (b) that this could be due to the inherent variability in LuCaP 23.1 tumor growth (which was not seen or mentioned by Thadani-Mulero et al)), and (c) that more in-depth follow up studies are needed to understand if slow and fast-growing variants differ in AR expression and drug-response.

4) Please correct the discussion of Nguyen et al., 2017, which is misrepresented on line 225. On page 668, end of first paragraph, Nguyen et al state: “The low dose of docetaxel showed efficacy only in LuCaP 86.2 (data not shown), reflecting a rare exceptional responder to docetaxel”. Thus, dosage is an unlikely reason for the failure of this line to respond in your hands.
 
5) Please explain in the text the various positive and negative control cells that were used in Figure 2. Include citations that demonstrated the expression of the relevant AR-isoforms.

Reviewer 1 ·

Basic reporting

No comment.

Experimental design

Tumor heterogeneity is frequently observed in animal studies, especially for those using fresh tumor tissues rather than cultured cell lines that has been adapted to uniform in vitro environment. Therefore, tumors may evolve (or genetically drift) individually, and growth pattern would be expected to be very diverse among them. Based on this concept, there are two major issues in the experimental design of this study.

First, the biomarkers or the targeted molecules need to be examined in every passage for quality assurance. In this study, different spliced forms of AR should be examined (in mRNA AND protein levels) at least in two stages: tumor expansion and implantation for randomization. This has been suggested in the previous comment but was not implemented in the current study.

Second, in such case, simple comparison in average tumor sizes between two groups cannot provide clear information, due to the standard deviation range will be too big in every time point of measurement.

Possibly because of the wide range of the data point distribution, the authors excluded the data of the fastest growing tumors from Fig. 1. This is by any way inappropriate. A better way of group comparison is Kaplan-Meier analysis. This will require a well-defined endpoint, which is totally lacking in the presented animal studies. One of the available solutions is to compare the growth rate constant of individual tumors (http://clincancerres.aacrjournals.org/content/early/2011/02/06/1078-0432.CCR-10-1762.short). The reviewer suggests to include it in the current study.

Validity of the findings

Since the expression of AR-v7 was not detected in LuCap23.1 by Western blot, and mRNA expression was not examined, no conclusion in the chemoresistant effect of AR-v7 can be drawn from this study. Without well-defined endpoint(s), the effect of docetaxel on LuCap23.1 PDX was also inconclusive.

Additional comments

Studying reproducibility should not be limited only on repeating the protocol of the study to be examined. The causes of variation need to be investigated, too. For such purpose, appropriate quality control/assurance step must be developed, and standard criteria (e.g. endpoint in animal study) should be included.

Reviewer 2 ·

Basic reporting

no comment

Experimental design

no comment

Validity of the findings

no comment

Additional comments

The authors have nicely addressed both reviewers' concerns throughout the paper and have added a concluding paragraph.

---

## Round 0.3 · Major Revisions

Based on our conversation and agreement by e-mail, please revise the manuscript to address point 2 as previously requested, i.e. remove the drug treatment data and revise the text accordingly.

In addition, please ascertain that the Discussion explains that the replication study failed to replicate the original results in three ways: (a) the cell line in question did not grow in a comparable way to the control, (b) the Arv7 variant was not detectable, and (c) the antibody used in the original study did not provide comparable data. These are important observations, which your study should report at this stage.

Please try to return the manuscript as soon as possible, given that no further experimentation is necessary.

Thank you!

---

## Round 0.4 · Minor Revisions

As discussed during our zoom meeting yesterday, please amend the discussion to clearly state that (a) the lack of response to docetaxel could also be due to a compromised quality of the reagent, and (b) a positive quality control for the reagent is missing because it was not factored into the plan at the stage of the Registered Report (since it was unexpected that these cells would not be responsive).

Further, please clarify to the readers why a higher dose (10 mg/kg) was used instead of the 5 mg/kg in the original publication.

Lastly, Thadani-Mulero et al. state in the Methods that treatment began at 200 mm3 tumor size. According to their data in Figure 6A and Suppl. Fig.4, treatment indeed started at 100 mm3. Your protocol is correct, but a clarifying note on this could be added in your Methods section.

---

## Round 0.5 · accepted · Accept

Thank you for the efforts in participating in this adventure, and the constructive interactions during this journey from Registered Report to Replication Study!

#